# CGXplain: Rule-Based Deep Neural Network Explanations Using Dual Linear Programs

**Konstantin Hemker, Zohreh Shams & Mateja Jamnik**
Department of Computer Science & Technology
University of Cambridge
Cambridge, UK
`{kh701, zs315, mj201}@cam.ac.uk`

## Abstract

Rule-based surrogate models are an effective and interpretable way to approximate a Deep Neural Network's (DNN) decision boundaries, allowing humans to easily understand deep learning models. Current state-of-the-art decompositional methods, which are those that consider the DNN's latent space to extract more exact rule sets, manage to derive rule sets at high accuracy. However, they a) do not guarantee that the surrogate model has learned from the same variables as the DNN (alignment), b) only allow optimising for a single objective, such as accuracy, which can result in excessively large rule sets (complexity), and c) use decision tree algorithms as intermediate models, which can result in different explanations for the same DNN (stability). This paper introduces **C**olumn **G**eneration e**X**plainer to address these limitations – a decompositional method using dual linear programming to extract rules from the hidden representations of the DNN. This approach allows optimising for any number of objectives and empowers users to tweak the explanation model to their needs. We evaluate our results on a wide variety of tasks and show that CGX meets all three criteria, by having exact reproducibility of the explanation model that guarantees stability and reduces the rule set size by $>80\%$ (complexity) at improved accuracy and fidelity across tasks (alignment).

## 1 Introduction

In spite of state-of-the-art performance, the opaqueness and lack of explainability of DNNs has impeded their wide adoption in safety-critical domains such as healthcare or clinical decision-making. A promising solution in eXplainable Artificial Intelligence (XAI) research is presented by global rule-based *surrogate models*, that approximate the decision boundaries of a DNN and represent these boundaries in simple IF-THEN-ELSE rules that make it intuitive for humans to interact with (Zilke et al., 2016; Shams et al., 2021). Surrogate models often use *decompositional* approaches, which inspect the latent space of a DNN (e.g., its gradients) to improve performance, while *pedagogical* approaches only utilise the inputs and outputs of the DNN.

In pursuit of the most accurate surrogate models, recent literature has primarily focussed on improving the fidelity between the DNN and the surrogate model, which refers to the accuracy of the surrogate model when predicting the DNN's outputs $\hat{y}$ instead of the true labels $y$. While state-of-the-art methods achieve high fidelity (Contreras et al., 2022; Espinosa et al., 2021), there are several qualitative problems with these explanations that hinder their usability in practice and have been mostly neglected in previous studies. First, if features are not fully independent, there is no guarantee that a surrogate model has learned from the same variables as the DNN, meaning that the surrogate model may provide misleading explanations that do not reflect the model's behaviour (**alignment**). Second, most rule extraction models optimise for the accuracy of the resulting rule set as a single objective, which can result in excessively large rule sets containing thousands of rules, making them impractical to use (**complexity**). Third, existing decompositional methods use tree induction to extract rules, which tends to be unstable and can result in different explanations for the same DNN, sometimes leading to more confusion than clarification (**stability**).

This paper introduces CGX (Figure 1) – a flexible rule-based decompositional method to explain DNNs at high alignment and stability, requiring only a fraction of the rules compared to current state-of-the-art methods. We combine and extend two recent innovations of decompositional explanations (i.e., using information from the hidden layers of the DNN) (Espinosa et al., 2021) and rule induction literature (i.e., generating boolean rule sets for classification) (Dash et al., 2018).

First, we suggest a paradigm shift for rule-based surrogate explanations that goes beyond optimising for accuracy as a single objective, allowing users to tailor the explanation to their needs. Concretely, we formulate the objective function of the intermediate model penalises the predictive loss as well as the number of rules and terms as a joint objective. Additionally, CGX allows to easily introduce further objectives. Second, we use a column generation approach as intermediate models, which have proven to be more accurate and stable than tree induction and other rule mining methods. Third, our algorithm introduces *intermediate error prediction*, where the information of the DNN's hidden layers is used to predict the error of the pedagogical solution (Equation 1). Fourth, we reduce the noise created by adding all rules from the DNN's latent representation by a) conducting direct *layer-wise substitution*, which reduces error propagation of the recursive substitution step used in prior methods and b) dismisses rules that do not improve the performance of the explanation model. This also reduces the need to choose between decompositional and pedagogical methods, since CGX converges to the pedagogical solution in its worst case performance.

CONTRIBUTIONS

- *Quality metrics*: We formalise three metrics (alignment, complexity, stability) that surrogate explanations need to achieve to be feasibly applied as an explanation model across datasets.

- *Alignment*: We improve alignment between the original and surrogate models, achieving 1-2% higher fidelity of the rule-based predictions and 10-20% higher Ranked Biased Overlap (RBO) of ranked feature importance representations.

- *Complexity*: We reduce the size of the rule sets used to explain the DNN, achieving rule sets with >80% less terms compared to state-of-the-art decompositional baselines.

- *Stability*: Our explanations are guaranteed to produce identical explanations for the same underlying model.

- *Decompositional value*: We demonstrate that decompositional methods are particularly useful for harder tasks, while pedagogical methods are sufficient for simple tasks.

## 2 RELATED WORK

**XAI & Rule-based explanations** XAI research has the objective of understanding *why* a machine learning model makes a prediction, as well as *how* the process behind the prediction works (Arrieta et al., 2020). This helps to increase trustworthiness (Floridi, 2019), identifying causality (Murdoch et al., 2019), as well as establishing confidence (Theodorou et al., 2017), fairness (Theodorou et al., 2017), and accessibility (Adadi & Berrada, 2018) in model predictions. *Global explainability methods* attempt to learn a representation that applies to every sample in the data, instead of only individual samples or features (local), and then provide a set of generalisable principles, commonly referred to as a *surrogate model* (Arrieta et al., 2020). Surrogate models can be either pedagogical or decompositional (Islam et al., 2021). **Pedagogical methods** train an explainable model on the predictions of the DNN $\hat{y}$ instead of the true labels $y$, still treating keep treating the DNN as a black-box (Confalonieri et al., 2020; Saad & Wunsch II, 2007). Pedagogical methods have a faster runtime since they ignore the latent space of the DNN, but sacrifice predictive performance (Zilke et al., 2016). **Decompositional methods** inspect the model weights or gradients and can therefore learn a closer representation of *how* the model makes a prediction at the expense of runtime.

One promising category of global decompositional methods are rule extraction models such as DeepRED (Zilke et al., 2016), REM-D (Shams et al., 2021), ECLAIRE (Espinosa et al., 2021), and DeXIRE (Contreras et al., 2022). These methods learn a set of conjunctive (CNF) or disjunctive normal form (DNF) rules $R_{x \mapsto \hat{y}}$ that approximate the neural network's predictions $\hat{y}$ (Zilke et al., 2016). Existing decompositional methods often use decision tree algorithms, such as C5.0 (Pandya & Pandya, 2015), for intermediate rule extraction. Thus, they learn rules that represent the relationship

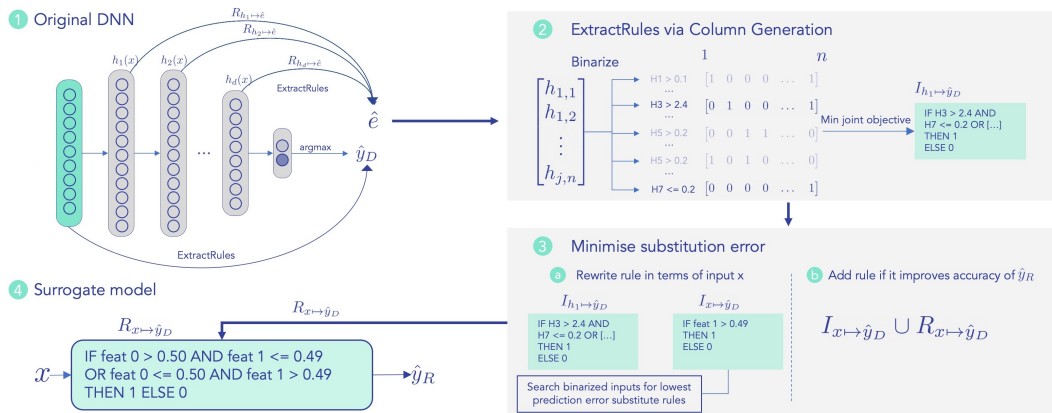

Figure 1: Overview of the decompositional `CGX` algorithm, showing the process to get from the DNN as starting point (1) to the explanation model (4) that approximates the DNN's decision boundaries. We 1(a) extract the rule set $R_{x \mapsto \hat{y}_D}$ by training an intermediate model on the DNN's predictions, and 1(b) on the error of that initial ruleset for each hidden layer. Our intermediate extraction through Column generation (2) allows optimising for multiple objectives to extract short and concise rule sets. The substitution step (3) rewrites the intermediate rules $I_{h_j \mapsto \hat{y}_D}$ in terms of the input variables $I_{x \mapsto \hat{y}_D}$ and adds them to the surrogate model (4) if they increase its fidelity.

between each hidden layer and the DNN predictions $R_{h_i \mapsto \hat{y}}$, which are then recursively substituted to be rewritten in terms of the input features as $R_{x \mapsto \hat{y}}$ (Shams et al., 2021). While existing surrogate methods achieve high fidelity, the resulting rule set $R$ is often still too large (thousands of rules) to clarify the model's behaviour in practice. Recent research has attempted to reduce the complexity of rule-based surrogates by running different decision tree algorithms, pruning methods (Shams et al., 2021), or clause-wise substitution (Espinosa et al., 2021). However, existing rule-based surrogate algorithms are heavily dependent on tree-based models used for rule generation. Thus, the performance is significantly sacrificed if the tree depth is too heavily restricted, despite reducing the size of the rule set.

**Rule induction methods** Another approach to explainability is to use explainable-by-design models, one of which are rule-based representations. Many of these methods use rule mining which first produces a set of candidate terms and then implements a rule selection algorithm which selects or ranks the rules from that search space. The problem with this is that the search space is inherently restricted (Lakkaraju et al., 2016; Wang et al., 2017). Another class of methods, such as RIPPER (Cohen, 1995) construct their rule sets by greedily adding the conjunction that explains most of the remaining data. This approach comes with the problem that the rule sets are not guaranteed to be globally optimal and commonly result in large rule sets. Two popular state-of-the-art rule induction methods that aim to control rule set complexity are Bayesian Rule Sets (BRS) (Wang et al., 2017) and Boolean rules from Column Generation (CG). BRS use probabilistic models with prior parameters to construct small-size DNF rule sets. Column generation uses binarisation and large linear programming techniques to efficiently search over the exponential number of possible terms, where the rule set size can be restricted with a complexity constraint in the objective function. While all of the above rule induction methods could be used for the rule extraction, we chose CG due to its stability and flexible formulation of the objective function.

## 3 METHODOLOGY

### 3.1 QUALITY METRICS

To improve on the shortcomings of existing decompositional methods, we first provide formal definitions to measure alignment, complexity, and stability. We assume an original model $f(x)$ (DNN) with $i$ hidden layers $h_i$ and the rule-based surrogate model $g(f(x))$ consisting of the rule set $R_{x \mapsto \hat{y}}$ that was extracted using an intermediate model $\psi(\cdot)$.

We define **complexity** as the size of the explanation rule set $|R_{x \mapsto \hat{y}}|$, expressed as the sum of the number of terms of all rules in $R$, i.e., $\min |R_{x \mapsto \hat{y}}|$.

We measure **alignment** between $f_x$ and $g_x$ in two different ways. First, we look at the ***performance alignment*** as fidelity, which measures the predictive accuracy of the model predictions $\hat{y}_g$ agains the original model predictions $\hat{y}_f$ as $\mu_{f,g} = \frac{1}{n} \sum_1^{n-1} (\hat{y}_f = \hat{y}_g)$. Second, we assess the ***feature alignment*** of the resulting explanations. Feature importance is a commonly used to understand which variables a model relies on when making predictions, represented as a ranked list. To validate that $f_x$ and $g_x$ are well-aligned, we want to ensure that both models rely on the same input features from $X$ in their predictions. Assuming two ranked lists $S$ and $T$, we calculate the Ranked Biased Overlap $\varphi_{ST}$ (Webber et al., 2010) as $\max \varphi(S, T, p) = \max(1 - p) \sum_{d=1} p^{d-1} A_d$, where $A_d$ is the ratio of list overlap size at depth $d$ and $w_d$ is the geometric progression $w_d = (1 - p)p^{d-1}$, a weight vectors used to calculate the weighted sum of all evaluation depths.

Finally, we define **stability** as rule sets that are identical on repeated calls of the explanation methods with the same underlying model. We run the explanation model $g_x$ on different seeds $s = \{0, 1, 2..., j\}$, where we want to ensure that the rule sets are equivalent as $R_{x \mapsto \hat{y}}(s_1) = R_{x \mapsto \hat{y}}(s_2)$.

### 3.2   COLUMN GENERATION AS INTERMEDIATE MODEL

We hypothesise that the majority of the complexity, stability, and alignment issues stem from the choice of the intermediate model $\psi(\cdot)$ in state-of-the-art decompositional methods. We use an adapted version of the column generation solver outlined in Dash et al. (2018). Instead of using $\psi(\cdot)$ as a standalone model, we will show that the column generation solver is well-suited as intermediate model in decompositional methods instead of commonly used tree-based algorithms such as C4.5/C5.0 Zilke et al. (2016); Shams et al. (2021). We start with the original restricted Master Linear Program which formulates from Dash et al. (2018) the Hamming loss, which counts the number of terms that have to be removed to classify the incorrect sample correctly. The Hamming loss is bound by an error and complexity constraint. We update the negative reduced cost of the pricing subproblem from (Dash et al., 2018) to include the hyperparameters for the number of rules ($\lambda_0$) and the number of terms ($\lambda_1$), which are linked to the complexity constraint as a dual variable. This formulation also makes it simple to add further parameters to the complexity constraint and negative reduced cost (e.g, adding a constraint that penalises rules or terms for only one particular class).

### 3.3   CG EXPLAINER

ECLAIRE outperforms other decompositional methods on fidelity, rule set size, and run time. Using column generation instead of tree induction as the intermediate model $\psi(\cdot)$, we reformulate the ECLAIRE algorithm as shown in Algorithm 1 with the core objective of improving the three quality metrics we set out. We introduce two versions of the column generation explainer – a pedagogical (`CGX-ped`) and a decompositional implementation (`CGX-dec`).

**CGX-ped** extracts rules from the intermediate model to predict the DNN predictions $\hat{y}_D$. This method ignores the latent space of the DNN, but can still outperform standalone column generation by guidance of the DNN's predictions:

$$\hat{y}_{ped} = R_{x \mapsto \hat{y}_D}(X) = \psi(X, \hat{y}_D) \tag{1}$$

**CGX-dec** (Algorithm 1) introduces three key innovations over other decompositional methods. First, we do not start with an empty rule set, but uses the pedagogical solution (Equation 1) as a starting point (line 2). Second, building on the pedagogical rule set, the algorithm iterates through the hidden layers. To improve on the pedagogical solution at each layer, we run *intermediate error prediction* by extracting rules by applying the intermediate model $\psi(\cdot)$ to predict the prediction error of the pedagogical solution $\hat{e}$ from each hidden layer (line 5). That is, we specifically learn rules that discriminate between false and correct prediction of the current best rule set, therefore resulting in rules that would improve this solution. The final update is the substitution method – previous approaches recursively replace the rules (Shams et al., 2021) or terms (Espinosa et al., 2021) of the hidden layer $h_{j+1}$ with the terms for each output class from the previous layer $h_j$ until all hidden rules can be rewritten in terms of the input features $X$. Since not every hidden layer can be perfectly represented in terms of the input, the substitution step always contains an error which propagates

---

**Algorithm 1** CGX-dec

---

**Input**: DNN $f_\theta$ with layers $\{h_0, h_1, ..., h_{d+1}\}$
**Input**: Labelled Training data $X = \{x^{(1)}, ..., x^{(N)}\}; Y = \{y^{(1)}, ..., y^{(N)}\}$
**Output**: Rule set $R_{x \mapsto \hat{y}}$

  1: $\hat{y}^{(1)}, ..., \hat{y}^{(N)} \leftarrow \arg\max(h_{d+1}(x^{(1)})), ..., \arg\max(h_{d+1}(x^{(N)}))$
  2: $R_{x \mapsto \hat{y}} \leftarrow \psi(X, \hat{y})$
  3: **for** hidden layer $i = 1, ..., d$ **do**
  4:     $x'^{(1)}, ..., x'^{(N)} \leftarrow h_i(x^{(1)}), ..., h_i(x^{(N)})$
  5:     $\hat{e} \leftarrow (\hat{y}_{ped} \neq \hat{y}_{nn})$
  6:     $R_{h_i \mapsto \hat{e}} \leftarrow \psi(\{(x'^{(1)}, \hat{e}_1), ..., (x'^{(N)}, \hat{e}_N)\})$
  7:     **for** rule $r \in R_{h_i \mapsto \hat{e}}$ **do**
  8:         $s \leftarrow \texttt{substitute}(r)$
  9:         $I_{x \mapsto \hat{y}} \leftarrow s \cup R_{x \mapsto \hat{y}}$
 10:         **if** $fid(\hat{y}_I, \hat{y}) > fid(\hat{y}_R, \hat{y})$ **then**
 11:             $R_{x \mapsto \hat{y}} \leftarrow I_{x \mapsto \hat{y}} \cup R_{x \mapsto \hat{y}}$
 12:         **end if**
 13:     **end for**
 14: **end for**
 15: **return** $R_{x \mapsto \hat{y}}$

---

down the layers as the same method is applied recursively. Instead, we use the direct rule substitution step outlined in Algorithm 2. Similar to the CG solver, we first binarise our input features as rule thresholds (line 1). After computing the conjunctions of the candidate rules, we calculate the error for each candidate and select the set of candidate terms with the lowest error (Algorithm 2, line 3) compared to the hidden layer predictions ($\hat{y}_{h_{ij}}$). Knowing that the substitution step still contains an error, some rules contribute more to the performance than others (rules with high errors are likely to decrease predictive performance). Therefore, the last update in Algorithm 1 is that the substituted rules resulting after the substitution step are only added to the rule set if they improve the pedagogical solution (lines 9 & 10).

## 4 EXPERIMENTS

Given the alignment, complexity, and stability shortcomings of existing methods, we design computational experiments to answer the following **research questions**:

- **Q1.1 Performance alignment**: Does the proven higher performance of column generation rule sets lead to higher fidelity with the DNN?

- **Q1.2 Feature alignment**: How well do aggregate measures such as feature importance from the rule set align with local explanation methods of the DNN?

- **Q2 Complexity**: Can we control the trade-off between explainability (i.e., low complexity) and accuracy by optimising for a joint objective?

---

**Algorithm 2** Direct rule substitution

---

**Input**: rule $r_{h_{ij} \mapsto \hat{y}}$
**Input**: Training data $X = \{x^{(1)}, ..., x^{(N)}\}$
**Hyperparameter**: # of rule candidate combinations $k$
**Output**: substituted rule(s) $r_{x \mapsto \hat{y}}$

  1: $X_{bin} \leftarrow \texttt{BinarizeFeatures}(X, bins)$
  2: $r_{cand} \leftarrow \texttt{ComputeConjunctions}(k, X_{bin})$
  3: $Errors_{r_{cand}} \leftarrow 1 - \frac{1}{N} \sum^{N-1} (\hat{y}_{h_{ij}} = \hat{y}_{r_{cand}})$
  4: $r_{x \mapsto \hat{y}} \leftarrow min(Errors_{r_{cand}})$
  5: **return** $r_{x \mapsto \hat{y}}$

---

- **Q3 Stability**: Do multiple runs of our method produce the same rule set for the same underlying model?

- **Q4 Decompositional value**: Is the performance gain of decompositional methods worth the higher time complexity compared to pedagogical methods?

## 4.1 Baselines & Setup

We use both pedagogical and decompositional explanation baselines in our experiments. For pedagogical baselines, we re-purpose state-of-the-art rule induction and decision tree methods to be trained on the DNN predictions $\hat{y}$ instead of the true labels $y$. Concretely, we use the C5.0 decision tree algorithm (Pandya & Pandya, 2015), Bayesian Rule Sets (Wang et al., 2017), and RIPPER (Cohen, 1995). As decompositional baselines, we implement the ECLAIRE algorithm as implemented in (Espinosa et al., 2021) which has been shown to outperform other decompositional methods in both speed and accuracy. Additionally, we benchmark against the standalone Column Generation method (Dash et al., 2018) trained on the true labels $y$ to show the benefit of applying it as an intermediate model in both pedagogical and decompositional settings. We run all baselines and our models on five different real-world and synthetic classification datasets, showing the scalability and adaptability to different numbers of samples, features, and class imbalances (Appendix A.1).

We run all experiments on five different random folds to initialise the train-test splits of the data, the random initialisations of the DNN as well as random inputs of the baselines. All experiments were run on a 2020 MacBook Pro with a 2GHz Intel i5 processor and 16 GB of RAM. For running the baselines, we use open-source implementations published in conjunction with RIPPER, BRS, and ECLAIRE, running hyperparameter search for best results as set out in the respective papers. For comparability, we use the same DNN topology (number and depth of layers) as used in the experiments in (Espinosa et al., 2021). For hyperparameter optimisation of the DNN, we use the `keras` implementation of the Hyperband algorithm (Li et al., 2018) to search for the optimal learning rate, hidden and output layer activations, batch normalisation, dropout, and L2 regularisation. The `CGX` implementation uses the MOSEK solver (Andersen & Andersen, 2000) as its `cvxpy` backend. The code implementation of the `CGX` algorithm can be found at `https://github.com/konst-int-i/cgx`.

## 5 Results

**Performance alignment (Q1.1)** The primary objective of performance alignment is the fidelity between the predictions of the rule set $\hat{y}_R$ compared to the model predictions $\hat{y}_{DNN}$, since we want an explanation model that mimics the DNNs behaviour as closely as possible. The results in Table 1 show that `CGX-ped` has a higher fidelity compared to the baseline methods on most datasets by approximately 1-2% whilst having significantly fewer rules. While RIPPER has a slightly higher fidelity on the MAGIC dataset, both `CGX-ped` and `CGX-dec` achieve competitive performance whilst only requiring 5% of the rules. This table also shows that a high fidelity does not guarantee a high accuracy on the overall task, which is visible on the FICO dataset. While `CGX` achieves a very high fidelity in this task, the overall accuracy is relatively low. This is caused by the underlying DNN struggling to perform well in this task. Notably, the performance of `CGX-dec` and `CGX-ped` is equivalent on the XOR dataset, indicating that there were no rules to add from the intermediate layers. This is because the XOR dataset is a relatively simple synthetic dataset, where the pedagogical version already identifies nearly the exact thresholds that were used to generate the target (see Figure 2(c)).

**Feature alignment (Q1.2)** Going beyond fidelity, the feature alignment score $\psi$ in Figure 2(a) shows the mean RBO score $\psi$ between the feature importance derived from the CGX rule set and the aggregated importance of local methods (SHAP and LIME) of the original DNN. A higher score shows that the two ranked lists are more aligned and, as such, the DNN and the rule-based surrogate model rely on the same features for their explanations more closely. Figure 2(a) compares the decompositional `CGX-dec` method to the best-performing decompositional baseline (ECLAIRE) and shows that `CGX-dec` achieves a higher feature alignment across all datasets compared to the baseline.

**Complexity (Q2)** Table 1 shows that both the pedagogical and decompositional methods achieve highly competitive results with only a fraction of the rules required. Compared to pedagogical baselines, `CGX-ped` outperforms on the majority of the tasks. While the pedagogical BRS baseline

Table 1: Overview of `CGX-ped` and `CGX-dec` performance alignment (fidelity) and complexity (# terms) compared to the baselines across datasets. `CGX-ped` outperforms all baselines across the majority of tasks. While RIPPER has a slightly higher fidelity on the MAGIC dataset, `CGX` only requires ~5% of the terms.

| DATASET | MODEL | RULE FID. | RULE ACC. | # RULES | # TERMS |
|---------|-------|-----------|-----------|---------|---------|
| XOR | CG (STANDALONE) | 78.0 ± 16.8 | 81.1 ± 18.5 | 5.2 ± 1.9 | 21.6 ± 12.7 |
|  | RIPPER (PED) | 53.5 ± 3.9 | 53.8 ± 4.0 | 7.4 ± 3.6 | 14.4 ± 7.5 |
|  | BRS (PED) | 91.3 ± 2.0 | 95.5 ± 1.3 | 9.0 ± 0.3 | 80.9 ± 3.0 |
|  | C5 (PED) | 53.0 ± 0.2 | 52.6 ± 0.2 | 1 ± 0 | 1 ± 0 |
|  | ECLAIRE (DEC) | 91.4 ± 2.4 | 91.8 ± 2.6 | 87 ± 16.2 | 263 ± 49.1 |
|  | CGX-PED (OURS) | **92.4 ± 1.1** | **96.7 ± 1.7** | **3.6 ± 1.8** | **10.4 ± 7.2** |
|  | CGX-DEC (OURS) | **92.4 ± 1.1** | **96.7 ± 1.7** | **3.6 ± 1.8** | **10.4 ± 7.2** |
| MAGIC | CG (STANDALONE) | 85.7 ± 2.5 | 82.7 ± 0.3 | 5.2 ± 0.8 | 13.0 ± 2.4 |
|  | RIPPER (PED) | **91.9 ± 0.9** | 81.7 ± 0.5 | 152.2 ± 14.6 | 462.8 ± 53.5 |
|  | BRS (PED) | 84.6 ± 2.1 | 79.3 ± 1.3 | 5.8 ± 0.3 | 24.1 ± 4.8 |
|  | C5 (PED) | 85.4 ± 2.5 | 82.8 ± 0.9 | 57.8 ± 4.5 | 208.7 ± 37.6 |
|  | ECLAIRE (DEC) | 87.4 ± 1.2 | 84.6 ± 0.5 | 392.2 ± 73.9 | 1513.4 ± 317.8 |
|  | CGX-PED (OURS) | 90.4 ± 1.7 | 80.6 ± 0.6 | **5.0 ± 0.7** | **11.6 ± 1.9** |
|  | CGX-DEC (OURS) | 91.5 ± 1.3 | 84.4 ± 0.8 | 7.4 ± 0.8 | 11.6 ± 1.9 |
| MB-ER | CG (STANDALONE) | 92.1 ± 1.1 | 92.0 ± 1.1 | 5.0 ± 0.7 | 15.4 ± 2.2 |
|  | RIPPER (PED) | 86.5 ± 2.2 | 85.2 ± 3.0 | 22.0 ± 9.2 | 30.2 ± 21.6 |
|  | BRS (PED) | 90.9 ± 1.2 | 88.4 ± 0.9 | 8.9 ± 1.1 | 57.6 ± 18.5 |
|  | C5 (PED) | 89.3 ± 1 | 92.7 ± 0.9 | 21.8 ± 3 | 72.4 ± 14.5 |
|  | ECLAIRE (DEC) | **94.7 ± 0.2** | **94.1 ± 1.6** | 48.3 ± 15.3 | 137.6 ± 24.7 |
|  | CGX-PED (OURS) | 93.7 ± 1.1 | 92.0 ± 0.9 | **4.2 ± 0.4** | **17.0 ± 1.9** |
|  | CGX-DEC (OURS) | **94.7 ± 0.9** | 92.4 ± 0.7 | 5.9 ± 1.1 | 21.8 ± 3.4 |
| MB-HIST | CG (STANDALONE) | 88.5 ± 2.3 | 91.1 ± 1.4 | 4.0 ± 0.7 | 19.4 ± 2.4 |
|  | RIPPER (PED) | 86.7 ± 3.7 | 88.1 ± 3.3 | 13.8 ± 3.4 | 35.0 ± 11.6 |
|  | BRS (PED) | 81.7 ± 2.1 | 79.9 ± 2.5 | 5.1 ± 0.2 | 40.3 ± 5.8 |
|  | C5 (PED) | 89.3 ± 1 | 87.9 ± 0.9 | 12.8 ± 3.1 | 35.2 ± 11.3 |
|  | ECLAIRE (DEC) | 89.4 ± 1.8 | 88.9 ± 2.3 | 30 ± 12.4 | 74.7 ± 15.7 |
|  | CGX-PED (OURS) | 89.1 ± 3.6 | 89.4 ± 2.5 | **5.2 ± 1.9** | **27.8 ± 7.6** |
|  | CGX-DEC (OURS) | **89.6 ± 3.6** | **90.2 ± 2.5** | 6.8 ± 2.0 | 32.2 ± 8.3 |
| FICO | CG (STANDALONE) | 86.4 ± 2.8 | 70.6 ± 0.4 | 3.3 ± 1.1 | 8.6 ± 3.6 |
|  | RIPPER (PED) | 88.8 ± 2.8 | 70.2 ± 1.0 | 99.2 ± 14.5 | 307.4 ± 41.6 |
|  | BRS (PED) | 84.8 ± 2.3 | 65.4 ± 2.1 | 3.1 ± 0.2 | 18 ± 3.2 |
|  | C5 (PED) | 72.7 ± 2.1 | 81.8 ± 1.6 | 34.8 ± 4.1 | 125.6 ± 35.2 |
|  | ECLAIRE (DEC) | 66.5 ± 2.5 | **84.9 ± 1.7** | 161.0 ± 12.3 | 298.0 ± 21.2 |
|  | CGX-PED (OURS) | 91.1 ± 0.1 | 70.5 ± 0.8 | **3.6 ± 1.1** | **9.6 ± 3.6** |
|  | CGX-DEC (OURS) | **92.4 ± 0.2** | 71.4 ± 1 | 5.1 ± 1.3 | 13.4 ± 2.1 |

produces fewer rules for some datasets (FICO and MB-HIST), their total number of terms are more than double those of `CGX` across all datasets due to longer chained rules being produced by this method. Additionally, the BRS fidelity is not competitive with `CGX-ped` or `CGX-dec`. Looking at ECLAIRE as our decompositional baseline, the results show that `CGX-dec` only requires a fraction of the terms compared to ECLAIRE. In the case of the Magic dataset, ECLAIRE required >100x more rules than our method, while for other datasets, the multiple ranges from 10-20x more rules required.

**Stability (Q3)** Figure 2(c) shows that `CGX` (both versions) results in identical explanations when running only the explainability method on a different random seed, keeping the data folds and random seed of the DNN identical. We observe that `CGX` produces the exact same rule set on repeated runs, while our decompositional baseline produces different explanations, which can be confusing to users. Note that this stability is different from the standard deviation shown in Table 1, where we would expect variation from different splits of the data and random initialisations of the DNN.

**Value of decomposition (Q4)** We acknowledge that the time complexity of decompositional methods scales linearly to the number of layers, which makes the pedagogical `CGX-ped` implementation an attractive alternative for very deep network topologies. To help decide whether to use pedagogical or decompositional methods, we looked at how much the information from the DNN's latent space

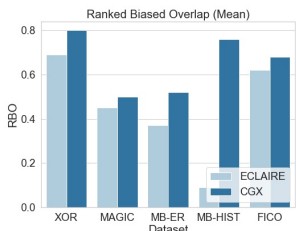
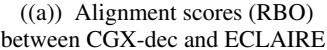

((a)) Alignment scores (RBO) between CGX-dec and ECLAIRE

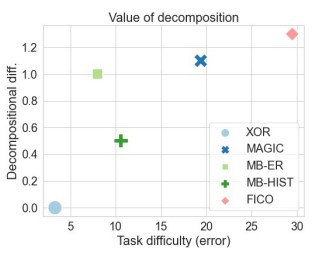
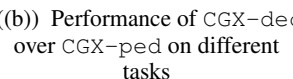

((b)) Performance of CGX-dec over CGX-ped on different tasks

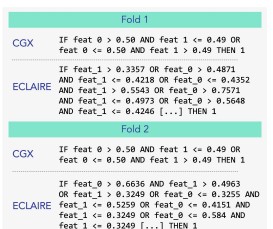

((c)) Ruleset Stability across folds

Figure 2: Overview of CGX performance with respect to alignment (a), task difficulty (b), and stability (c). Subfigure (a) shows the mean Ranked Biased Overlap of CGX compared to ECLAIRE shows that CGX's rule set show a higher *feature alignment*. Subfigure (b) is comparing task difficulty (DNN prediction error, x-axis) with the incremental fidelity improvement (y-axis) when using CGX-dec over CGX-ped. As tasks get more difficult, using the CGX-dec adds relatively more fidelity compared to the CGX-ped. Subfigure (c) shows that CGX has exact reproducibility for the same underlying model.

(lines 3-10 in Algorithm 1) improves the pedagogical solution (line 2 in Algorithm 1). Figure 2(b) shows that the added performance gained from information of the hidden layers is related to the difficulty of the task. For "easy" tasks, (i.e., those where the DNN has a high accuracy/AUC such as the XOR task), CGX-ped and CGX-dec converge to the same solution, since no rules from the hidden layers increase the fidelity. Figure 2(b) shows that the performance difference increases with the difficulty of the task. For the FICO task, where the DNN accuracy is only just over 70%, the surrogate model gains the most information from the hidden layers.

## 6 DISCUSSION

This paper introduces a global decompositional method that uses column generation as intermediate models. We improve rule-based explanations by intermediate error predictions from the latent space of a DNN, coupled with layer-wise substitution to reduce error propagation. CGX enables research and industry to customise surrogate explanations for different end users by parameterising the accuracy-explainability trade-off. First, we introduced a quantitative measure to analyse the **feature alignment** between the surrogate model and local explanations of the DNN and show that our surrogate model explanations are more closely aligned to other local explanation methods of the original model. Second, the design of the objective functions allows assigning a higher cost to surrogate model complexity (i.e., the number of terms in the rule set) using an extra hyperparameter. We demonstrate that this achieves significantly **lower complexity** and enables users to control the accuracy-interpretability trade-off by setting higher or lower penalties on the number of rules. Third, the results show that CGX is independent of its initialisation (solution to the Master Linear Program), which leads to **improved stability** compared to methods using tree induction for rule extraction. Additionally, CGX requires **fewer hyperparameters** compared to tree-based algorithms such as C5, hence requiring less fine-tuning to achieve competitive results. While this introduces the lambda parameter to enable users to control the length of the resulting rule set, it is also possible to run the solver unconstrained. Beyond these benefits, having rule-based surrogate models enables end users to *intervenability* by users, as they can amend the rule set to encode further domain knowledge.

The key limitation of CGX and decompositional methods more generally is that the runtime is highly dependent on the number of hidden DNN layers and the number of columns in $X$. We attempt to mitigate this problem by showing that CGX-ped is a highly competitive alternative, especially for simple tasks. For more difficult tasks, however, the decompositional method still delivers better explanations with higher fidelity. The implementation will be open-sourced as a pip-installable Python package.

ACKNOWLEDGMENTS

KH acknowledges support from the Gates Cambridge Trust via the Gates Cambridge Scholarship.

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

# A DATASETS

## A.1 DATASETS

**MAGIC** is a particle physics dataset used to simulate the registration of either high-energy gamma particles or background hadron cosmic radiation based on imaging signals from a ground-based atmostpheric telescope. It has 19k samples with $\sim$35% in the minority class and 10 features derived from the "shower image" of the pulses left by the incoming photons Bock et al. (2004).

**Metabric-ER** predicts the Immunohistochemical subtypes of 1980 patients using 1000 features including tumour characteristics, clinical traits, gene expression patterns, and survival rates. In this dataset, $\sim$24% of patients are Estrogen-Receptor-positive (ER), meaning that these tumours have ERs that allow the tumour to grow.

**Metabric-Hist** contains 1004 mRNA expressions of 1694 patients to predict two of the most common histological subtypes of breast cancer tumours, where positive diagnoses make up 8.7% of the samples – Invasive Lobular Carcinoma (ICL) or Invasive Ductal Carcinoma (IDC) Pereira et al. (2016).

**XOR** is a synthetic dataset used as a common baseline to evaluate the performance of rule extractors. The dataset generates a supervised dataset with 10 features of the form $(x_j^{(i)}, y_i)_{i=1}^{1000}$ where every data point in $x_i \in [0,1]^{10}$ is independently sampled from a uniform distribution. The binary labels $y_i$ are assigned by XOR-ing the result of the rounded first two dimensions $y_i = round(x_1^{(i)}) \bigoplus round(x_2^{(i)})$.

**FICO** is a finance dataset originally designed for an explainable ML challenge containing home equity line of credit (HELOC) applications by homeowners with the task of predicting whether the applicant repays their HELOC credit within 2 years. The dataset contains 10,459 applicants with 24 features on the spending habits of each applicant as well as an external risk estimate of each.

