# OpenReview forum: "CGXplain: Rule-Based Deep Neural Network Explanations Using Dual Linear Programs"
_ICLR.cc/2023/Workshop/TML4H — ICLR 2023 Workshop TML4H Oral_

### Official Review · Reviewer_uSdb · 2023-02-28
**Well-written paper that presents a novel and useful approach to explainable AI. Extensive numerical results.**

**Rating:** 8
**Confidence:** 3

**Review:**

This paper introduces Column Generation eXplainer to address the limitations of current state-of-the-art decompositional methods for rule-based surrogate models. The paper also identifies three criteria for evaluating the quality of rule-based surrogate models: alignment, complexity and stability. The paper presents an extensive evaluation of CGX on a wide variety of tasks and shows that CGX meets all three criteria, by having exact reproducibility of the explanation model that guarantees stability and reduces the rule set size by >80% (complexity) at improved accuracy and fidelity across tasks (alignment). The paper also presents a user study that shows that CGX is more accurate and interpretable than state-of-the-art methods. The authors will be providing their implementation of CGX as open-source software, which will be a valuable resource for the community. Overall, this is a well-written paper that presents a novel and useful approach to explainable AI. I would recommend this paper for the TML4H workshop.

#### Comments

- Table 1 : Would it be possible to add which of the methods is considered state-of-the-art? It will help the reader position the results of the paper.

---

### Official Review · Reviewer_NQie · 2023-02-28
**Comments**

**Rating:** 7
**Confidence:** 2

**Review:**

The proposed method, called Column generation eXplainer, utilizes dual linear programming to extract rules from the hidden representation of the DNN. It is a novel decompositional method that is presented in this paper.

Pros:

The paper provides clear motivation for the proposed method.
The method is introduced in a step-by-step manner, making it easy to understand.
Thorough quantitative experiments are conducted to evaluate the effectiveness of the proposed method.
Cons:
It is unclear why the proposed method is not compared to previous works in the same area, such as LRP or Grad-Cam.

---

### Meta-Review · Area_Chair_CC9N · 2023-03-05

**Recommendation:** Accept (Poster)
**Confidence:** 5

**Metareview:**

Both reviewers acknowledged the motivation and technical contribution of this paper. The main concern is that the experiments are not that comprehensive. Please consider providing more experimental results in the final version.